# Gut Microbiota, Inflammatory Bowel Disease, and Cancer: The Role of Guardians of Innate Immunity

**DOI:** 10.3390/cells12222654

**Published:** 2023-11-19

**Authors:** Vincenzo Giambra, Danilo Pagliari, Pierluigi Rio, Beatrice Totti, Chiara Di Nunzio, Annalisa Bosi, Cristina Giaroni, Antonio Gasbarrini, Giovanni Gambassi, Rossella Cianci

**Affiliations:** 1Institute for Stem Cell Biology, Regenerative Medicine and Innovative Therapies (ISBReMIT), Fondazione IRCCS “Casa Sollievo della Sofferenza”, 71013 San Giovanni Rotondo, Italy; v.giambra@operapadrepio.it (V.G.); b.totti@operapadrepio.it (B.T.); c.dinunzio@operapadrepio.it (C.D.N.); 2Medical Officer of the Carabinieri Corps, Health Service of the Carabinieri General Headquarters, 00197 Rome, Italy; danilo.pagliari@gmail.com; 3Department of Translational Medicine and Surgery, Fondazione Policlinico Universitario A. Gemelli IRCCS, Catholic University of Rome, 00168 Rome, Italy; pierluigi.rio18@gmail.com (P.R.); antonio.gasbarrini@unicatt.it (A.G.); giovanni.gambassi@unicatt.it (G.G.); 4Department of Medicine and Technological Innovation, University of Insubria, via H Dunant 5, 21100 Varese, Italy; annalisa.bosi@uninsubria.it (A.B.); cristina.giaroni@uninsubria.it (C.G.)

**Keywords:** toll-like receptors, NOD-like receptors, inflammatory bowel diseases, colorectal cancer, microbiota, genetic polymorphisms

## Abstract

Inflammatory bowel diseases (IBDs) are characterized by a persistent low-grade inflammation that leads to an increased risk of colorectal cancer (CRC) development. Several factors are implicated in this pathogenetic pathway, such as innate and adaptive immunity, gut microbiota, environment, and xenobiotics. At the gut mucosa level, a complex interplay between the immune system and gut microbiota occurs; a disequilibrium between these two factors leads to an alteration in the gut permeability, called ‘leaky gut’. Subsequently, an activation of several inflammatory pathways and an alteration of gut microbiota composition with a proliferation of pro-inflammatory bacteria, known as ‘pathobionts’, take place, leading to a further increase in inflammation. This narrative review provides an overview on the principal Pattern Recognition Receptors (PRRs), including Toll-like receptors (TLRs) and NOD-like receptors (NLRs), focusing on their recognition mechanisms, signaling pathways, and contributions to immune responses. We also report the genetic polymorphisms of TLRs and dysregulation of NLR signaling pathways that can influence immune regulation and contribute to the development and progression of inflammatory disease and cancer.

## 1. Introduction

Inflammatory bowel diseases (IBDs) are characterized by a persistent inflammatory status that leads to chronic mucosal damage and consequently to an increased risk of cancer development. Although IBD-associated colorectal cancer (CRC) accounts for approximately 2% of all CRC, this complication is related to a significant increase in the mortality rate in IBD patients [1]. The correlation between IBD and CRC has been well established, but the underlying specific mechanisms are still not completely understood. Several factors are implicated in this pathogenetic pathway, such as chronic inflammation, the innate and adaptive immune system, gut microbiota, environmental agents, and xenobiotics. The close link between inflammation and cancer can be considered as a milestone of modern medicine: it was first described by Rudolf Virchow over 150 years ago and then confirmed by several scientific studies [2,3]. Chronic inflammation is associated with about 20% of all human cancers, and the coexistence of long-standing inflammation and cancer development is particularly valid in CRC [4].

In IBD, a persistent chronic inflammatory status heavily affects the gut mucosa homeostasis; at this level, a complex interplay between the immune system and gut microbiota occurs [5]. Several conditions are implicated in the development of persistent mucosal damage, and the most important factor is represented by the alteration in the gut permeability. In the condition of ‘leaky gut’, a disequilibrium between the immune system and gut microbiota (GM) exists and several inflammatory pathways are activated, leading to the proliferation of the pro-inflammatory T-cell subsets, such as T helper Th1, Th2, and Th17, with a corresponding downregulation of the anti-inflammatory ones, such as T regulatory cells (Tregs) [5]. The pro-inflammatory adaptive immune cells are responsible for the secretion of several pro-inflammatory cytokines and other mediators able to recruit, at the gut mucosa, pro-inflammatory cells contributing to mucosal damage [6]. In this scenario of impaired gut mucosa homeostasis, gut microbiota composition is altered, and pro-inflammatory bacteria, known as ‘pathobionts’, proliferate and further enhance the immune response. This complex pathological mechanism, involving both the immune system and GM, is driven by the presence of specific receptors on immune cells, able to be activated by microbial components. Pattern Recognition Receptors (PRRs) are a family of receptors that recognize pathogen-associated molecular patterns (PAMPs), present on pathogens, and danger-associated molecular patterns (DAMPs), released during tissue damage. They act as sentinels of the immune system and can drive innate immune responses against infectious agents [7]. This narrative review provides an overview of the principal PRRs, including Toll-like receptors (TLRs) and NOD-like receptors (NLRs), discussing their recognition mechanisms, signaling pathways, and contributions to immune responses. We describe genetic variations in TLRs and the dysregulation of NLR signaling pathways that can influence immune responses and contribute to the development and progression of inflammatory disease and cancer. 

## 2. Genetic, Immune, and Environmental Factors in Inflammatory Bowel Diseases 

Inflammatory bowel diseases (IBDs) are chronic and relapsing inflammatory disorders classified as Crohn’s disease (CD), ulcerative colitis (UC), and undetermined colitis. In the 21st century, IBDs have emerged as global diseases characterized by a rapidly changing epidemiology worldwide, although a consolidated theory suggests that the pathogenesis is strictly dependent upon environmental exposure associated with society’s Westernization [8]. Indeed, the incidence of IBDs is highest in Western countries, where, however, it has plateaued at the beginning of the twenty-first century, while it is gradually becoming more common in newly industrialized countries in Latin America and Asia [8]. 

CD is a chronic IBD characterized by inflammation that can affect any part of the gastrointestinal tract. It is a complex and multifactorial disorder with both genetic and environmental contributions. Genetic studies have identified several susceptibility loci associated with CD, including the nucleotide-binding oligomerization domain (NOD)2/caspase activation and recruitment domain (CARD)15, the autophagy-related 16 like 1 (ATG16L1), and interleukin (IL)23 receptor [9,10,11]. These genetic variants play a role in the regulation of innate immune responses, autophagy, and the balance of pro-inflammatory and anti-inflammatory cytokines. Environmental factors, such as diet, smoking, and microbial dysbiosis, also contribute to the pathogenesis of CD. Dysregulation of the mucosal immune response leads to excessive inflammation, tissue damage, and the characteristic clinical features of CD, including abdominal pain, diarrhea, and weight loss. 

UC involves the rectum and colon and is characterized by chronic inflammation and ulceration of the colonic mucosa. Similar to CD, UC is a complex disorder influenced by both genetic and environmental factors. Genome-wide association studies have identified genetic loci associated with UC, including genes involved in immune regulation, such as IL23R, IL10, and human leukocyte antigen (HLA) genes [12,13,14]. Environmental factors, such as smoking, diet, and the gut microbiome, also contribute to the pathogenesis of UC. Dysregulation of the mucosal immune response leads to an uncontrolled inflammatory cascade, resulting in the characteristic symptoms of UC, including bloody diarrhea, abdominal pain, and urgency. 

The pathogenesis of IBD is extremely complex and multifactorial. Current treatment approaches for IBD aim to induce and maintain remission, minimize disease-related complications, and improve patients’ quality of life. Treatment options include immune modulators, such as corticosteroids, immunomodulatory drugs (e.g., thiopurines), and biologic agents targeting specific cytokines (e.g., antitumor necrosis factor (TNF) antibodies). Personalized medicine approaches are also being explored, considering individual genetic profiles, disease phenotypes, and therapeutic response predictors. Emerging therapies targeting novel pathways, such as Janus kinase (JAK) inhibitors, cytokine blockers, and gut microbiome-based interventions, offer promising avenues for future treatment options [15]. 

The pathogenesis of IBD involves an imbalance in the immune response, characterized by the excessive activation of pro-inflammatory pathways and inadequate regulation of anti-inflammatory mechanisms. Numerous immune cell types and signaling molecules contribute to this dysregulated immune response. For instance, in CD, the increased production of pro-inflammatory cytokines, such as TNF-α, IL-1β, and IL-6, perpetuates chronic inflammation [16,17]. In UC, the aberrant activation of immune cells, including T-cells, macrophages, and dendritic cells, leads to sustained mucosal inflammation [18]. The intricate interplay between immune cells, cytokines, and molecular pathways contributes to the complex immune dysregulation observed in IBD.

Innate immune dysfunction plays a significant role in the pathogenesis of IBD; in fact, dendritic cells, macrophages, and epithelial cells are fundamental in sensing and responding to microbial components through PRRs, like TLRs and NLRs. Dysregulation of these innate immune pathways can lead to an excessive immune response and contribute to chronic intestinal inflammation. Studies have shown the altered expression and function of TLRs and NLRs in IBD patients, suggesting innate immune abnormalities [19,20]. For example, impaired TLR signaling, particularly through TLR4 and TLR2, has been associated with defective immune responses and increased susceptibility to intestinal inflammation [21,22]. 

Epigenetic modifications, including DNA methylation, histone modifications, and non-coding RNA regulation, have emerged as critical players in the pathogenesis of IBD. These modifications can influence gene expression patterns and alter immune responses in the intestinal mucosa. Studies have revealed aberrant DNA methylation patterns in IBD patients, affecting key genes involved in immune regulation, epithelial barrier function, and inflammatory pathways [23,24]. Histone modifications, such as acetylation and methylation, also play a role in IBD pathogenesis by modulating the gene expression and chromatin structure [25]. Furthermore, the dysregulation of non-coding RNAs, including microRNAs and long non-coding RNAs, has been implicated in IBD development and progression [26,27]. 

Lastly, several dietary factors could influence the inflammatory pathways in IBD and cancer. Diet can modulate the symptoms of IBD both in a direct and indirect manner, for example, through the modulation of GM [28]. Moreover, bacterial metabolites may modulate pro- and anti-inflammatory pathways [29]. IBD patients develop weight loss, sarcopenia, malnutrition, and obesity [30]. In particular, it is well known that different fat sources may differently dysregulate the immune system and tumor progression [31]. Furthermore, diet phenolic compounds are able to modulate lipid metabolism and oxidative stress, by positively modulating gut microbiota, with an increase in the amount of Bacteroidetes and reduced Firmicutes. In this way, assuming the phenolic compound through a whole grain diet could prevent hyperlipidemia and related inflammatory and cancer diseases [32]. In IBD patients, diet is important not only during the exacerbation of disease but also to prevent selective malnutrition and obesity.

## 3. Primary Sensors of the Innate Immune System

### 3.1. Toll-like Receptors (TLRs)

TLRs play a critical role in the innate immune system, acting as primary sensors for the detection of PAMPs, including bacterial lipopolysaccharides, viral nucleic acids, and fungal components [33].

These type 1 transmembrane proteins act as receptors and are expressed on various immune cells, including macrophages, dendritic cells, and B cells; they activate signaling pathways leading to the production of pro-inflammatory cytokines and type I interferons [34]. 

To date, 11 and 12 functional TLRs have been identified in humans and mice, respectively [35]. Each TLR is specific for different PAMPs of bacteria, viruses, fungi, and parasites. These include lipoproteins (recognized by TLR1, TLR2, TLR6, and TLR10), viral double-stranded (ds) RNA (TLR3), lipopolysaccharide (LPS) (TLR4), flagellin (TLR5), microbial single-stranded (ss) RNA (TLR7 and TLR8), and DNA (TLR9) [36]. Moreover, intracellular TLRs (TLR 3, 7, 8, 9) are involved in antiviral immunity, while transmembrane TLRs (TLR 1, 2, 4, 5, 6, 10) in the immune response against extracellular pathogens [37].

After ligand binding, TLRs activate specific signaling pathways, mainly through the Myeloid differentiation primary response 88 (MyD88)-dependent or Toll/IL-1R domain-containing adaptor-inducing IFN-beta (TRIF)-dependent pathways. The MyD88 pathway is involved in the production of pro-inflammatory cytokines, while the TRIF one is responsible for the induction of type I interferons [38]. TLRs’ activation results in the production of cytokines, chemokines, and interferons, which play crucial roles in inflammation, immune cell recruitment, and antimicrobial response. TLRs also enhance antigen presentation, promoting the activation of adaptive immune responses [39].

### 3.2. TLR Aberrant Signaling and Polymprphisms

Aberrant TLR signaling has been associated with several diseases, including infections, autoimmune disorders, and cancer. Moreover, the dysregulation of TLRs can lead to either excessive inflammation or impaired immune responses, contributing to disease pathogenesis and progression [40] (Figure 1).

Genetic variations in TLR genes, referred to as polymorphisms, have been extensively studied to understand their impact on immune responses and disease susceptibility. 

For example, regarding infectious diseases, polymorphisms of TLR genes have been associated with altered susceptibility to various pathogens [37,41] and with selection pressure caused by different pathogens acting in populations of different ethnicities [42]. Moreover, TLR7 and 8 genes are both situated on the X chromosome. For this reason, it is possible to hypothesize a different immune response between men and women; moreover, polymorphisms of these two TLRs can show a different expression in the immune response between the two sexes [43].

Recently, it has been shown that the L412F polymorphism in TLR3 is linked to severe COVID-19 disease due to a poor recognition of the SARS-CoV-2 virus [36], through its ability to alter the three-dimensional conformation of the C-terminal leucine-rich repeat (LRR) domain, affecting the interaction between the TLR3 and its ligand [44]. Moreover, this polymorphism is also associated with a higher level of pro-inflammatory cytokines and co-stimulatory molecules [45]. The Q11L is a polymorphism of intracellular TLR7 that can reduce the ability to recognize other viral agents, such as Hepatitis C virus and human immunodeficiency virus [46].

Due to the impairment in activity of recognizing pathogens, the I602S and N248S variants of TLR1 have been associated with several infectious diseases, such as tuberculosis, malaria, and leprosy [47]. Other different polymorphic variants of TLR 8 and 9 are also linked to tubercular infections [48]. In the same way, the S249P polymorphism of TLR6, on the one hand, could represent a risk factor, when expressed in homozygosis, for tuberculosis in African-Americans, but, on the other hand, may protect against asthma [37]. TLR2 polymorphisms are able to reduce the immune responses against parasites; indeed, the R677W and R753Q variants decrease the pathogen recognition function with a consequent downregulation of the transcription factor nuclear factor kappa-light-chain-enhancer of activated B cells (NF-kB) activity and pro-inflammatory cytokines production [37,49]. The TLR5 R392STOP polymorphism can reduce the immune response to flagellin [50]; moreover, it may transfer resistance to a complex autoimmune disease, such as systemic lupus erythematosus (SLE) [51]. Patients harboring the D299G and T399I variants of TLR4 are associated with a hyporesponsiveness to LPS [52]. Furthermore, D299G is associated with inflammatory diseases, such as IBD [53].

The Asp299Gly polymorphism of TLR4 has been implicated in the pathogenesis of systemic lupus erythematosus (SLE) and rheumatoid arthritis (RA) [54,55]. This polymorphism affects the function of TLR4, leading to the impaired recognition of microbial components, which may contribute to dysregulated immune responses and to the development of autoimmunity [56]. Similarly, TLR2 polymorphisms have been associated with an increased susceptibility to autoimmune diseases, such as type 1 diabetes (T1D) [57]. Recently, the rs352140 polymorphism of TLR9 has been associated with T1D too and represents a risk factor for susceptibility to this disease [58].

Taken together, this evidence shows that TLR polymorphisms play an important role not only in infectious diseases, but also in the pathogenesis of autoimmune diseases and they could be potential genetic markers for disease susceptibility.

The identification of TLR polymorphisms associated with autoimmune diseases has significant implications for personalized medicine and therapeutic targeting. Understanding the impact of these genetic variations on immune responses can help stratify patients based on their genetic risk profiles and tailor treatment strategies accordingly. For example, patients with TLR polymorphisms associated with hyperactive immune responses may benefit from targeted immunomodulatory therapies to restore immune homeostasis. Modulating TLR activation through small-molecule inhibitors or biologic agents may help attenuate excessive immune responses and ameliorate disease manifestations. 

TLR polymorphisms have also been associated with cancer susceptibility and progression [59]. Genetic polymorphisms in TLRs can affect the recognition of tumor-associated antigens and the subsequent activation of antitumor immune responses [59]. For instance, TLR4 polymorphisms have been linked to an increased risk of developing some types of cancer, including colorectal and gastric cancer [59]. In IBD, the associations with TLR gene polymorphisms have to consider the gene–gene and gene–environmental interactions. In a recent meta-analysis, TLR1 rs5743611, TLR4 rs4986790, TLR4 rs4986791, TLR6 rs5743810, and TLR9 rs352140 polymorphisms have been shown to represent genetic biomarkers of IBD in specific ethnicities [60] (Table 1).

## 4. NOD-like Receptors (NLRs)

In humans, NOD-like receptors (NLRs) are a family of intracellular PRRs that play a crucial role in innate immunity [76,77]. These receptors present a nucleotide-binding oligomerization domain (NOD), an N-terminal effector-binding one, and a C-terminal leucine-rich repeat (LRR) domain [77]. NLRs can be classified into five subfamilies, depending on the N-terminal domain: NLRA or Class II transactivator (CIITA) characterized by an acidic domain, NLRBs or neuronal apoptosis inhibitor proteins (NAIPs) with a baculovirus inhibitor repeats (BIRs), NLRCs with a caspase-recruitment domain (CARD), NLRPs with a pyrin domain (PYD), and NLRX1 localized to the mitochondria with a CARD-related X effector domain [78]. After ligand binding, the LRR presents a conformational change, which exposes the N-terminal domain [79]. Through their unique structure, NLRs recognize various PAMPs and DAMPs, initiating signaling cascades that lead to immune activation [76] and host defense against pathogens [80]. In fact, NLRs, such as NLR family pyrin domain containing (NLRP)1, NLRP3, and NLRP6, activating the inflammatory protease caspase-1 are referred to as inflammasomes. NLRs such as NOD1 and NOD2, act as cell death inducers, transcriptional activators, promoters of NF-κB and mitogen-activated protein kinase (MAPK) pathways, and regulators of type I interferons [81]. 

### 4.1. Dysregulation of NLR Signaling

Dysregulation of NLR signaling pathways contributes to chronic inflammation and plays a role in the pathogenesis of several diseases. Genetic polymorphisms in NLRs can have profound functional consequences on immune responses, increasing disease susceptibility, thus contributing to the development of inflammatory disorders. For instance, the single nucleotide polymorphisms (SNPs) can disrupt the normal function of NLR proteins, leading to altered immune signaling pathways and impaired host defense mechanisms. The NOD2 polymorphisms have been associated with an increased risk of IBD, by dysregulating immune activation and inducing chronic inflammation. Common mutations, such as Leu1007fsinsC, Arg702Trp, and Gly908Arg, have been described in association with ileal involvement and strictures in Crohn’s disease [82]. 

PAMPs and DAMPs stimulate the molecular assembly of inflammasomes, cytosolic complexes constituted by multimeric proteins such as NLRs. These complexes recruit pro-caspase-1, whose subsequent activation as caspase-1 promotes the production of pro-inflammatory cytokines, as well as a type of cell death called pyroptosis [83,84]. The NLRP3 inflammasome, a well-studied complex, plays a crucial role in the host immune response against pathogens; however, its dysregulation leads to disease. For instance, it is over-expressed in advanced CRC with a worse prognosis [85]. Moreover, genetic variations in NLRP3 are linked to autoinflammatory disorders. For example, SNP rs35829419 has been associated with increased NLRP3 inflammasome activation and the pathogenesis of cryopyrin-associated periodic syndromes (CAPS), characterized by recurrent fever and systemic inflammation [86,87]. In CAPS patients with hyperactive NLRP3 variants, targeted therapies inhibiting the NLRP3 inflammasome or neutralizing excessive pro-inflammatory cytokines have shown promising results in clinical trials [88,89]. Since the NLRP3 inflammasome is also involved in the pathogenesis of IBD [90], in the last few years, the idea of targeting it as a therapeutic strategy in these inflammatory disorders has gained increasing attention [83]. 

One approach for targeting NLRs is the development of small-molecule inhibitors that selectively block NLR activation or downstream signaling cascades [91]. These inhibitors can interfere with the assembly and activation of the NLRP3 inflammasome, thereby reducing the production of pro-inflammatory cytokines, such as IL-1β and IL-18. Furthermore, therapeutic strategies aimed at enhancing the function of specific NLRs (e.g., NOD2) have been explored. For instance, the use of peptidoglycan-derived muramyl dipeptide (MDP) analogues can activate NOD2 signaling and restore immune responses in some diseases [92].

Moreover, the functional consequences of genetic polymorphisms in NLRs can shed light on disease mechanisms and allow us to identify potential therapeutic targets beyond the affected NLR itself. 

### 4.2. The Role of Caspase Activation and Recruitment Domain (CARD) Proteins

The downstream effector proteins of NLRs, such as Caspase Activation and Recruitment Domain (CARD) proteins, have also emerged as attractive therapeutic targets for the treatment of various inflammatory and autoimmune diseases [80] 

CARD proteins are a well-known protein interaction module found in a wide array of proteins typically involved in biochemical processes related to inflammation and apoptosis. The CARD has been well studied and has been shown to be associated with several human diseases including cancers, neurodegenerative diseases, and immune disorders [93].

CARD-containing proteins, allowing for the recruitment and activation of caspase-1 [94], play a crucial role in immune system activities, and their dysregulation leads to aberrant immune responses, contributing to the onset and progression of inflammatory disorders. NOD2 is a CARD-containing protein encoded by the NOD2/CARD15 gene, one of the most investigated IBD genes, since its mutations have been linked to an increased risk of developing Crohn’s disease [9]. NOD2 is involved in the recognition of bacterial components and the regulation of intestinal immune responses. Dysfunctional NOD2 signaling resulting from NOD2/CARD15 mutations can disrupt immune homeostasis in Crohn’s disease [72]. Furthermore, another deregulated CARD protein, such as CARD9, is implicated in the pathogenesis of inflammatory disorders, including fungal infections and IBD [69], contributing to the uncontrolled inflammation observed in these conditions. In healthy conditions, CARD9 and kinase SYK are involved in a protective inflammasome activation; however, polymorphisms in the CARD9 gene promote IBD and colon carcinogenesis [72,73].

The development of small-molecule inhibitors that selectively target CARD proteins or their downstream signaling components aims to disrupt CARD-mediated inflammasome assembly and activation, thereby reducing the release of pro-inflammatory cytokines and attenuating the inflammatory response [95]. By specifically targeting the CARD domains involved in protein–protein interactions and signaling cascades, these inhibitors offer the potential to modulate immune responses in a precise and controlled manner.

In addition to small molecules, biological agents such as monoclonal antibodies have been designed to target specific CARD proteins or downstream inflammatory cytokines. Monoclonal antibodies can neutralize the activity of CARD proteins or block the signaling cascades initiated by these, suppressing the excessive inflammation and restoring immune homeostasis [96].

As described above, the pivotal roles of NLRs and CARD proteins in immune regulation and disease pathogenesis make them attractive therapeutic targets [97], through promising strategies explored both in preclinical and clinical studies [97]. For instance, among the small-molecule inhibitors, MCC950 can inhibit the NLRP3 inflammasome activation and subsequent release of pro-inflammatory cytokines, offering potential therapeutic benefits in diseases like gout, CAPS, and IBD [91,98,99]. Moreover, antibody-based approaches, as well as gene-silencing techniques targeting NLRs and CARD proteins, are other potential therapeutic interventions [97]. 

## 5. Gut Microbiota in the Development of IBD and IBD-Associated Colorectal Cancer

The gut microbiota plays a crucial role in the pathogenesis of IBD and CRC by altering intestinal mucosa, microenvironmental homeostasis, and the mucosal immune system [100] (Table 2). Gut microbiota is composed of both beneficial bacteria contributing to gut mucosa homeostasis and harmful bacteria, known as pathobionts, responsible for gut inflammation and mucosal damage. Gut microbiota acts as a first-line defense against invading pathogens but it may also be responsible for the induction of the inflammatory response and the generation of genotoxic products derived from bacteria that increase cancer risk. Intestinal inflammation may be the result of a disturbance in the ratio between beneficial and harmful bacteria that results in a breakdown of the physiological gut homeostasis and consequent mucosal damage [101]. This alteration in the homeostatic composition and function of gut microbiota is also known as ‘intestinal dysbiosis’. Several pieces of evidence indicate that both IBD and CRC are associated with intestinal dysbiosis [4]. The inflammation related to the proliferation of pathobionts alters the normal composition of gut microbiota and results in intestinal barrier dysfunction. The state of ‘leaky gut’ favors the translocation of bacteria from the gut lumen to the *lamina propria* and the development of a strong inflammatory response due to the activation of TLRs and NF-κB pathways, responsible for the production of several pro-inflammatory cytokines and chemokines [33,102]. Experimental data have confirmed that TLR4-deficient mice have a decreased level of intestinal inflammation and TLR4 deficiency may prevent colitis-associated neoplasia [65]. On the other hand, TLR4 over-expression is associated with increased levels of inflammatory mediators, such as TNF-alpha, cyclooxygenase (COX)-2, and IL-12, with a higher susceptibility to both acute colitis and colitis-associated CRC [65].

It is now well ascertained that specific changes in the composition, number, and stability of gut microbiota are correlated to the development of intestinal diseases, including IBD and CRC. Although the microorganisms harboring the human gut are resilient to some unstable environmental conditions induced, for example, by diet modifications or drug treatment, the exposure to several factors, such as high-fat diet, drugs, smoke, age, and genetics, may profoundly change the bacterial composition and function, altering the symbiotic interplay with the host, thus contributing to the etiopathogenesis of IBD [115]. Indeed, microbial instability relates to IBDs and reflects both a reduction in the biodiversity and richness in the number of species within a community, which may be circumscribed to the inflamed regions in CD patients and relates to the disease activity index [116]. IBD patients harbor a less diverse gut microbiome characterized by a decrease in the abundance of bacteria belonging to Firmicutes and Bacteroidetes phyla and a relative growth of the Proteobacteria [117]. Although specific pathobionts have not been clear-cut identified yet, a recent study, aimed at investigating the gut microbiota imbalance at different taxonomic levels for healthy volunteers and CD and UC patients, shows bacterial groups which are altered in IBD patients do not co-exist well with common commensal gut bacteria, whereas bacterial groups which did not change in IBD patients were found to commonly co-exist with commensal gut microbiota [117].

Several reports suggest that increased levels of *Bacteroides fragilis*, *Escherichia coli*, *Enterococcus faecalis*, and *Streptococcus bovis* are pathogenic for IBD and are related to the development of CRC [4]. In particular, the presence of *Bacteroides fragilis* and *Escherichia coli* is related to tissue damage consequent to the activation of the pro-inflammatory and tumorigenic transcription factor signal transducer and activation of transcription (STAT)3 and the pro-inflammatory cytokine IL-17. Activation of STAT3 is related to several biochemical mechanisms influencing human diseases, both in inflammation and in cancer. In particular, persistent STAT3 activation in cancer cells is linked to cell survival, angiogenesis, and metastatic processes, contributing to increased inflammation and tumorigenesis [103]. *Enterococcus faecalis* is responsible for the production of the damaging reactive oxygen species (ROS), while *Streptococcus bovis* and *Bacteroides fragilis* are associated with the production of pro-inflammatory and pro-angiogenic cytokines, such as IL-6, IL-8, and IL-17 [104]. Moreover, an experimental model has demonstrated that IL-10 knock-out mice have a higher susceptibility to develop IBD and rectal dysplasia and cancer induced by *Enterococcus faecalis* [105]. Accordingly, the modification of the gut microbiota in IL-10 knock-out mice by probiotic Lactobacilli was related to a reduced prevalence of intestinal inflammation and CRC development [106]. Furthermore, microbial-driven IL-23 increases with inflammation and is able to promote the pro-inflammatory Th17 cells [118]. 

Intestinal infections may be related to the perpetuation of inflammation and increased risk of cancer. Episodes of *Salmonella*/*Campylobacter* gastroenteritis have been associated with an increased risk of developing IBD, and alterations in the TLR4 gene may predispose to these *Gram*-negative bacteria infections and generic increased susceptibility to enteric infections [107]. Thus, pathogenic infections may change the commensal composition of gut microbiota and disrupt commensal tolerance, leading to the IBD-related chronic inflammation and the consequent increased susceptibility to develop CRC [33].

On the other hand, several bacterial species within the *Clostridium*, *Lactobacillus*, *Faecalibacterium*, and *Bifidobacterium* genus seem to play a beneficial role against IBD [108]. For example, various Clostridia strains can promote Tregs development in colonic mucosa and thus may protect from colitis. Several studies correlate the depletion of the butyrate producer *Faecalibacterium prausnitzii* with CRC development [110,111]. In different in vivo models of chemically induced colitis, *Faecalibacterium prausnitzii* has been demonstrated to attenuate the severity of intestinal inflammation through the production of metabolites able to stimulate a tolerogenic cytokine profile, the enhancement of the intestinal barrier function, as well as the inhibition of NF-κB signaling and IL-8 production [109].

Due to the crucial role of *Faecalibacterium prausnitzii* in maintaining gut and host physiology, the microorganism has been proposed as a biomarker of CRC [109]. However, the protective effects of *Faecalibacterium prausnitzii* are not clear-cut defined since other studies have shown no depletion of the microorganism in CRC [112]. 

*Fusobacterium nucleatum* is suggested as another predictive and prognostic biomarker for CRC. In particular, the intratumoral presence of this CRC-enriching microorganism correlates with a poor prognosis due to a higher microsatellite instability and gene mutation [113]. Different experimental models demonstrated that *Fusobacterium nucleatum* promotes tumor progression, metastatization, and chemoresistance through its ability to influence tumor cells and several tumor microenvironment components (i.e., extracellular matrix, immune, and stromal cells). However, the association between *Fusobacterium nucleatum* and CRC initiation is also still unresolved and needs to be further elucidated [114]. *Fusobacterium nucleatum* cannot be completely considered as a pro-carcinogenic microorganism since its role depends upon the genetic background of the host, tumor microenvironment, and environmental factors.

## 6. The Interplay between TLR, Immunity, and Gut Microbiota in the Development of IBD and IBD-Associated Colorectal Cancer

TLRs and NOD, being the first outpost contrasting environmental enemies, may be considered as the most important mediators activating the pro-inflammatory response. TLRs and NOD, in fact, are particularly activated by pathobionts and govern the activation of a strong inflammatory response, leading to the transcription of various cytokines and chemokines [102]. 

Alteration in TLR signaling is also related to changes in microbiota composition and the disturbance of mucosal homeostasis. Among TLRs, TLR2 and TLR4 are mainly related to intestinal inflammation and their expression has been shown to be increased in the macrophages of IBD patients [119]. In this way, TLR2 signaling is strongly associated with the typical IBD mucosal damage, as TLR2 is able to modulate T-cell functions both directly, stimulating Th17 response [120], and indirectly, reducing the suppressive function of Tregs by promoting a shift toward IL-17 production [121]. Moreover, TLR4 has also been shown to be strongly connected to IBD pathogenesis. TLR4 is the first identified TLR in the mammalian system and is able to recognize the LPS of Gram-negative bacteria. In a healthy condition, TLR4 is expressed at a low level in intestinal epithelial cells (IECs). Its role is quite controversial; in fact, on the one hand, TLR4 is linked to beneficial actions, such as for the induction of an anti-inflammatory response providing protection from pathogens and promoting mucosal integrity, but, on the other hand, TLR4 may also cause tissue destruction and ulceration [63]. TLR4 has been demonstrated to be increased in IBD patients and it may be considered as an active participant in IBD disease development [122]. It was indeed reported that TLR2 and TLR4 expression is directly correlated with the *Enterococcus faecalis*, *Porphyromonas*, and *S. bovis*, and inversely, to the amount of *Lactobacillus*, Roseburia, and Bifidobacteria [64]. Interestingly, TLR immune signaling induced by microbiota is also implicated in cancer progression and the response to different conventional treatments [123,124,125]. Different studies have indeed reported that the transcriptional and protein levels of TLR4 are significantly high in the colonic mucosa of patients with colorectal cancer (CRC) compared with the controls [66,67] and increase with the disease progression [65,126]. Moreover, Luo et al. have recently shown in a mouse model that TLR9 may be associated with the development of CRC by regulating the NF-κB signaling pathway [127]. Nonetheless, other studies have reported that TLR9 agonists exerted an antitumor effect in CRC [39,40,85,86], suggesting that these findings are still controversial.

TLR polymorphisms have also been associated with cancer susceptibility and progression [59], as a TLR may also act as a modulator of CRC risk [4]. Genetic variations in a TLR can affect the recognition of tumor-associated antigens and the subsequent activation of antitumor immune responses [59]. As cited above, TLR4 polymorphisms have been linked to an increased risk of developing certain types of cancer, including CRC and gastric cancer [59]. Understanding the role of TLR polymorphisms in cancer pathogenesis can provide insights into the interplay between the immune system and tumor development. In this way, clinical data derived from a large study performed on CRC patients revealed that rs3775292, a specific TLR3 polymorphism, is significantly associated with rectal cancer, and the rs11536898 polymorphism of TLR4 is associated with colon cancer [62]. Notably, this study revealed that these associations were influenced by environmental factors, such as recent use of NSAIDs, cigarette smoking, dietary carbohydrates, and saturated fat [62]. Other evidence has shown that TLR4 may promote the development of colitis-associated tumors and it may also be associated with metastasis in CRC [62]. TLR4 expression has been shown in both normal colon mucosa and CRC cell lines, whereas a loss of its expression may be linked to increased metastases [68]. A large study conducted on a group of advanced and metastatic CRC revealed that specific TLR variants, such as TLR4 Asp299Gly, TLR4 Thr399Ile, TLR9 T1237C, TLR9 T1486C, and TLR2 -196 to -174 del/del homozygous genotypes, were significantly associated with CRC. Additionally, TLR4 Asp299Gly and Thr399Ile polymorphisms were significantly associated with concomitant Kirsten Rat Sarcoma (KRAS) gene mutations [61]. These polymorphisms are located in the encoding region of the TLR4 ectodomain and are related to a reduction in cytokine expression, contributing to an increased susceptibility to CRC [128,129].

NOD2 may also be associated with IBD and CRC. In fact, in 2004, a link between NOD2 polymorphisms and the risk of CRC was first described [130]. In this way, several studies confirmed a significantly higher incidence of NOD2 mutations in patients with CRC [4,75]. In particular, a large meta-analysis showed that some variants in NOD2, such as the R702W, G908R, and 3020insC variants, may be associated with a higher CRC susceptibility in Caucasians [70]. Although NOD2 may be considered as the major IBD susceptibility gene, its role in CRC is poorly defined. The pathogenetic pathway involving NOD2 in increasing the risk of developing CRC was clarified by Udden et al. In fact, these authors demonstrated that NOD2-deficient mice are highly susceptible to experimental colorectal tumorigenesis independent of gut microbial dysbiosis. In particular, NOD2, on the one hand, is able to activate the NF-kB and MAPK signaling axis in response to bacterial muramyl dipeptide (MDP), favoring the production of inflammatory mediators, and, on the other hand, it is also able to inhibit the TLR-mediated activation of the same NF-kB and MAPK pathways. NOD2 plays a crucial role in the suppression of inflammation and tumorigenesis in the colon by downregulating the TLR signaling pathways [69].

## 7. Conclusions

The intricate link among the gut microbiota, mucosal immune system, environment, and genetics plays a crucial role in the pathogenesis of IBD and CRC. Gut dysbiosis is associated with IBD and cancer and in individuals genetically predisposed to IBD, it can stimulate an aberrant immune response. Most importantly, genetic predisposition to IBD is associated with PRRs; indeed, genetic variations in TLRs and NLRs can trigger an inflammatory response by modifying microbial recognition patterns. Moreover, the microbiota can stimulate inflammasomes by prompting pro-inflammatory mediators of inflammation. Taking into account the gut microbiota composition and its metabolites, host signaling pathways, and genetic polymorphisms, the guardians of innate immunity, such as TLRs and NLRs, can trigger and perpetrate different inflammatory patterns and eventually cancer. Multiple mechanisms regulate TLR signaling to prevent excessive inflammation and maintain immune homeostasis. Targeting TLRs for therapeutic interventions has shown promise in various diseases, including sepsis, IBD, and cancer [131]. At the same time, NLR and CARD proteins play critical roles in innate immune signaling and disease pathogenesis. The intricate interactions between these components contribute to immune regulation and inflammatory responses. Understanding the mechanisms underlying NLR and CARD activation, the functional consequences of genetic variations, and their implications for disease susceptibility will pave the way for the development of new targeted therapies against these components of innate immune signaling. These therapies hold great potential for the treatment of inflammatory and autoimmune diseases. In conclusion, knowledge of specific PRRs genetic variations can guide the development of personalized treatment strategies.

## Figures and Tables

**Figure 1 cells-12-02654-f001:**
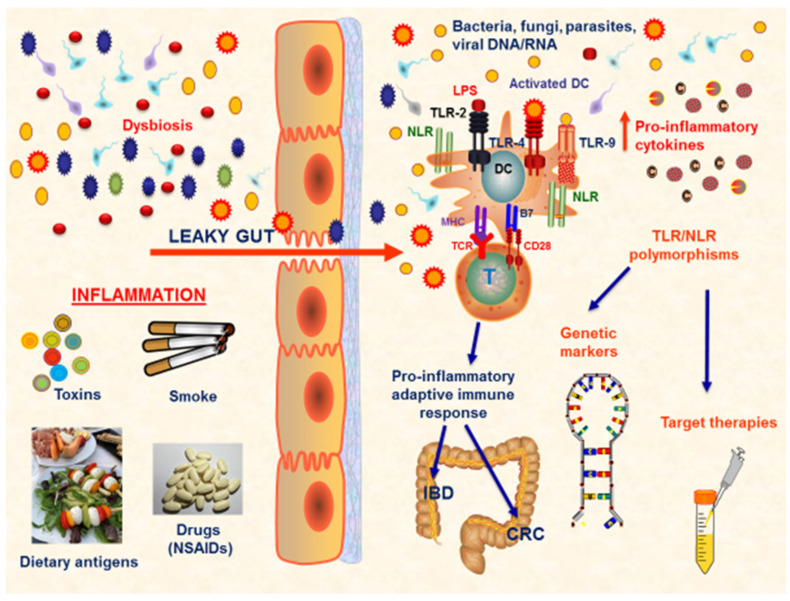
TLR response at gut mucosa in IBD and CRC. IBD and CRC are both characterized by the presence of a persistent chronic inflammatory status that heavily affects the gut mucosa homeostasis. Several conditions are implicated in the development of mucosal damage, such as dysbiosis, smoke, toxins, dietary antigens, and drugs. The most important factor is represented by the alteration in the gut permeability. In the condition of ‘leaky gut’, a disequilibrium between the immune system and microbiota exists and several inflammatory pathways are activated, leading to the proliferation of pro-inflammatory cells, such as Th1, Th2, and Th17 cells, with a corresponding downregulation of the anti-inflammatory Tregs, and the production of pro-inflammatory mediators. In this *scenario* of impaired gut mucosa homeostasis, the gut microbiota composition is altered, and pro-inflammatory bacteria, known as ‘pathobionts’, proliferate and further enhance the immune response. This complex pathological mechanism is driven by the presence of specific receptors on immune cells, able to be activated by microbial components, such as TLRs and NLRs. Several TLR/NLR polymorphisms have been evaluated to be linked to specific pathological patterns, both in IBD and CRC. Thus, these polymorphisms may be considered as disease-related genetic markers and may guide the development of personalized target therapies. Abbreviations: NSAIDs (non-steroidal anti-inflammatory drugs); TLR (Toll-like receptors); NLR (NOD-like receptors); DC (dendritic cell); IBD (inflammatory bowel disease); CRC (colorectal cancer); LPS (lipopolysaccharide).

**Table 1 cells-12-02654-t001:** Different PRRs, their ligands and functions in IBD and CRC.

Mediators	Ligands	Role in IBD	REF	Role in CRC	REF
TLR-2	Lipoproteins	- extracellular pathogen immunity; - mediates intestinal inflammation and its expression has been shown to be increased in macrophages of IBD patients; - modulations of T-cells functions both directly, stimulating Th17 response, and indirectly, reducing the suppressive function of Tregs by promoting a shift toward IL-17 production;	[33]	- TLR2 -196 to -174 del/del homozygous genotypes are significantly associated with CRC (***human study***);	[61]
TLR-3	Viral ds-RNA	- antiviral immunity;		- TLR3 rs3775292 polymorphism, is significantly associated with rectal cancer (***human study***);	[62]
TLR-4	LPS	- extracellular pathogen immunity; - causes tissue destruction and ulceration; - mediates intestinal inflammation and its expression is increased in macrophages of IBD patients; - over-expression of TLR4 is associated to increased inflammatory mediators, such as TNF-alpha, COX-2, and IL-12, with higher susceptibility to IBD; - alterations in the TLR4 gene may predispose to Gram-negative bacteria infections and generic increased susceptibility to enteric infections; - is increased in IBD patients and it may be considered as an active participant in IBD disease development; - TLR4 deficient mice have decreased level of intestinal inflammation (***animal study***); - TLR4 D299G and rs4986790 polymorphisms are associated with IBD (***animal study***);	[63] [64] [65] [52]	- over-expression of TLR4 is associated to increased inflammatory mediators, such as TNF-alpha, COX-2, and IL-12, with higher susceptibility to colitis-associated CRC (***human study***); - TLR4 rs11536898 polymorphism is associated with colon cancer (***human study***); - TLR4 Asp299Gly, TLR4 Thr399Ile homozygous genotypes are significantly associated with CRC (***human study***); - TLR4 Asp299Gly and Thr399Ile polymorphisms were significantly associated with concomitant KRAS gene mutations (***human study***); - TLR4 expression may be linked to increase metastases (***human study***); - TLR4 deficiency may prevent colitis-associated neoplasia (***animal study***);	[66] [67] [62] [61] [61] [68] [65]
TLR-6	Lipoproteins	- TLR6 rs5743810 polymorphism is associated with IBD (***human study***);	[60]	- TLR6 rs3796508 has a crucial role as a protective factor against colorectal cancer; - TLR6-deficient mice presented higher risk of developing cancer and a worse overall outcome;	[60]
TLR-9	Microbial ss-RNA	- antiviral immunity; - TLR9 rs352140 polymorphism is associated with IBD (***human study***);	[58]	- TLR9 T1237C and T1486C homozygous genotypes are significantly associated with CRC (***human study***);	[61]
NOD2	Bacterial peptidoglycan (PGN) fragments	- NOD2 activates the NF-kB and MAPK signaling axis in response to bacterial MDP favoring the production of inflammatory mediators (***animal study***); - Dysfunctional NOD2 signaling may contribute to the pathogenesis of Crohn's disease;	[69]	- NOD2 is able to suppress inflammation and tumorigenesis in the colon downregulating the TLR signaling pathways (***animal study***); - NOD2 polymorphisms increase the risk of CRC; - NOD2 R702W, G908R, and 3020insC variants may be associated in higher CRC susceptibility in Caucasians (***human study***);	[69] [70]
CARD9	Various PAMPs and DAMPs	- triggers the activation of NF-κB and STAT3, inducing the production of pro-inflammatory cytokines; - dysregulation of CARD9 signaling can disrupt immune responses and contribute to the uncontrolled IBD related inflammation;	[71]	- highly expressed in CRC tumor tissue; - highly expressed in tumor-infiltrating macrophages rather than cancer cells and is associated with CRC tumor metastasis and advanced histopathologic stage (***animal study***); - dysregulated CARD9 is a critical risk factor in the progression of CRC;	[72] [73]
CARD15	Various PAMPs and DAMPs	- CARD15 mutations are associated with higher susceptibility to CD (***animal study***); - mutations in the CARD15 gene have been linked to an increased risk of developing Crohn's disease.	[74]	- CARD15 mutations are associated with higher susceptibility to CRC; - the CARD15 R702W variant may be a predisposing factor to sporadic CRC (***human study***).	[75]

**Table 2 cells-12-02654-t002:** The role of gut microbiota in IBD and CRC.

Bacteria	Role in IBD	REF	Role in CRC	REF
*Bacteroides fragilis*, *Escherichia coli, Enterococcus faecalis,* and *Streptococcus bovis.*	- their levels are pathogenic for IBD.	[4]	- their levels are related to the development of CRC.	[4]
*Bacteroides fragilis* and *Escherichia coli*	- are related to tissue damage consequent to the activation of STAT3 and IL-17; - persistent STAT3 activation contributes to increase inflammation.	[103]	- persistent STAT3 activation in cancer cells is linked to cell survival, angiogenesis, and metastatic processes, contributing to increase tumorigenesis.	[103]
*Enterococcus faecalis*	- is responsible for the production of the ROS; - IL-10 knock-out mice have higher susceptibility to develop IBD induced by E*nterococcus faecalis* (animal model).	[104] [105]	- IL-10 knock-out mice have higher susceptibility to develop rectal dysplasia and cancer induced by E*nterococcus faecalis* (animal model);	[105]
*Streptococcus bovis* and *Bacteroides fragilis*	- are associated with the production of pro-inflammatory and pro-angiogenic cytokines, such as IL-6, IL-8, and IL-17.	[104]		
Lactobacilli	- modification of the gut microbiota in IL-10 knock-out mice by these bacteria was related to a reduced prevalence of intestinal inflammation (animal model).	[106]	- modification of the gut microbiota in IL-10 knock-out mice by probiotic was related to a reduced prevalence of CRC development (animal model).	[106]
*Salmonella/Campylobacter*	- episodes of *Salmonella/Campylobacter* gastroenteritis have been associated with increased risk of developing IBD (human study); - alterations in the TLR4 gene may predispose to these *Gram*-negative bacteria infections and generic increased susceptibility to enteric infections (human study).	[107] [107]		
*Clostridium*, *Lactobacillus*, *Faecalibacterium*, and *Bifidobacterium*	- their genus seems to play a beneficial role against IBD; - various Clostridia strains can promote Tregs development in colonic mucosa and thus may protect from colitis.	[108]		
*Faecalibacterium prausnitzii*	- has been demonstrated to attenuate the severity of intestinal inflammation through the production of metabolites able to stimulate a tolerogenic cytokine profile, the enhancement of the intestinal barrier function as well as the inhibition of NF-κB signaling and IL-8 production.	[109]	- several studies correlate the depletion of the butyrate producer *Faecalibacterium prausnitzii* with CRC development (human study); - due to its crucial role in maintaining gut and host physiology, the microorganism has been proposed as a biomarker of CRC; - its protective effects are not clear cut-defined since other evidences have shown no depletion of the microorganism in CRC (human study).	[110] [111] [109] [112]
*Fusobacterium nucleatum*			- is suggested as another predictive and prognostic biomarker for CRC; - the intratumoral presence of this CRC-enriching microorganism correlates with a poor prognosis due to higher microsatellite instability and gene mutation (human study); - promotes tumor progression, metastatization and chemoresistance through its ability to influence tumor cells and several tumor microenvironment components (human study); - the association between *Fusobacterium nucleatum* and CRC initiation is still unresolved and needs to be further elucidated; - cannot be completely considered as a pro-carcinogenic microorganism since its role depends upon the genetic background of the host, tumor microenvironment and environmental factors.	[113] [113] [114] [114]

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
