# Peer review of "Gut Microbiota, Inflammatory Bowel Disease, and Cancer: The Role of Guardians of Innate Immunity"

_cells, 2023, doi:10.3390/cells12222654_

Round 1

Reviewer 1 Report

Comments and Suggestions for Authors

This article discusses the main pattern recognition receptors (PRRs), including Toll-like receptors and NOD-like receptors, discusses their recognition mechanisms, signaling pathways and contributions to the immune response, and also describes the genetic variation of TLR and the dysregulation of NLR pathways, which may affect the immune response and facilitate the production of inflammatory diseases and cancers.The following are the shortcomings of the article:

1.     The introduction chapter begins without having two characters apartno empty rows between paragraphs in each section

2.     The Inflammation system with IBD should be introduced, please refer this reference (Cancer Research, 2020, 80(12): 2564-2574.).

3.     The title series is too few, the article hierarchy sense is weaksections 2 and 3 can be subdivided by one or two headings

4.     The abbreviation is mislocated and should be placed above the reference.

5.     The literature reference number annotation method is wrong, using the Chinese brackets and large numbers

6.     The font size of lines 107-121 is too small to coordinate with the full text

7.     The font format of the table in the text is not standard, and the font with smaller characters is used and written in the upper left corner of the table

8.     In this paper, there are too few schematic of metabolic process and biochemical recognition mechanism, which of vivid and intuitive expression

9.     The text spacing in lines 255 and 277 is too large and does not match with the text spacing of the full text

10.  “4. Genetic, Immune and Environmental Factors in Inflammatory Bowel Diseases”.

The diet infactors such as polyphenols should be consideration. Please refer this reference (Food & Function, 2022, 13(24), 12686-12696; Food Bioscience. 50(2022): 101946).

11.  The last reference is mis formed, the number does not require parentheses, and the document title size should be narrowed.

12.  The reference should be updated in recent years.

Comments on the Quality of English Language

This article discusses the main pattern recognition receptors (PRRs), including Toll-like receptors and NOD-like receptors, discusses their recognition mechanisms, signaling pathways and contributions to the immune response, and also describes the genetic variation of TLR and the dysregulation of NLR pathways, which may affect the immune response and facilitate the production of inflammatory diseases and cancers.The following are the shortcomings of the article:

1.     The introduction chapter begins without having two characters apartno empty rows between paragraphs in each section

2.     The Inflammation system with IBD should be introduced, please refer this reference (Cancer Research, 2020, 80(12): 2564-2574.).

3.     The title series is too few, the article hierarchy sense is weaksections 2 and 3 can be subdivided by one or two headings

4.     The abbreviation is mislocated and should be placed above the reference.

5.     The literature reference number annotation method is wrong, using the Chinese brackets and large numbers

6.     The font size of lines 107-121 is too small to coordinate with the full text

7.     The font format of the table in the text is not standard, and the font with smaller characters is used and written in the upper left corner of the table

8.     In this paper, there are too few schematic of metabolic process and biochemical recognition mechanism, which of vivid and intuitive expression

9.     The text spacing in lines 255 and 277 is too large and does not match with the text spacing of the full text

10.  “4. Genetic, Immune and Environmental Factors in Inflammatory Bowel Diseases”.

The diet infactors such as polyphenols should be consideration. Please refer this reference (Food & Function, 2022, 13(24), 12686-12696; Food Bioscience. 50(2022): 101946).

11.  The last reference is mis formed, the number does not require parentheses, and the document title size should be narrowed.

12.  The reference should be updated in recent years.

Author Response

Rome, 14th November, 2023

Dear Editor of “Cells”,

first of all, my coauthors and I would like to thank You sincerely for this opportunity of cooperation, following the submission of the paper “Gut microbiota, inflammatory bowel disease and cancer: the role of guardians of innate immunity” and its possible publication upon “Cells”.

We profoundly thank the reviewer for the comments and useful suggestions aimed at improving the final version of the paper.

This is a point-by-point list of changes made in the paper:

Reviewer #1

This article discusses the main pattern recognition receptors (PRRs), including Toll-like receptors and NOD-like receptors, discusses their recognition mechanisms, signaling pathways, and contributions to the immune response, and also describes the genetic variation of TLR and the dysregulation of NLR pathways, which may affect the immune response and facilitate the production of inflammatory diseases and cancers. The following are the shortcomings of the article:

  1. The introduction chapter begins without having two characters apart,no empty rows between paragraphs in each section

Thanks for the comment. We have modified the text accordingly

  1. The Inflammation system with IBD should be introduced, please refer this reference (Cancer Research, 2020, 80(12): 2564-2574.).

We have discussed the reference, as suggested.

  1. The title series is too few, the article hierarchy sense is weak;sections 2 and 3 can be subdivided by one or two headings

We have added title series and subdivided sections 2 and 3. We have changed the order of the paragraphs.

  1. The abbreviation is mislocated and should be placed above the reference.

Thanks for the comment. We have checked all the abbreviations and placed in the right place.

  1. The literature reference number annotation method is wrong, using the Chinese brackets and large numbers

We have modified the style of references, as requested.

  1. The font size of lines 107-121 is too small to coordinate with the full text

We have modified the font size of the original 107-121. We have used a smaller font size because it is the Figure 1 legend.

  1. The font format of the table in the text is not standard, and the font with smaller characters is used and written in the upper left corner of the table

We have modified the font format of the table, as suggested-

  1. In this paper, there are too few schematic of metabolic process and biochemical recognition mechanism, which of vivid and intuitive expression

We have added some metabolic processes, as suggested.

  1. The text spacing in lines 255 and 277 is too large and does not match with the text spacing of the full text

We have modified the text spacing, accordingly.

  1. “4. Genetic, Immune and Environmental Factors in Inflammatory Bowel Diseases”.

The diet infactors such as polyphenols should be consideration. Please refer this reference (Food & Function, 2022, 13(24), 12686-12696; Food Bioscience. 50(2022): 101946).

We have added discussion on diet compounds, as suggested

  1. The last reference is mis formed, the number does not require parentheses, and the document title size should be narrowed.

We have modified the references. Thank you.

  1. The reference should be updated in recent years.

We have updated several references in recent years.

We thank You for your constructive critique and we hope the review process has led to an improved manuscript.

If additional changes are warranted, we will make them.

We hope that this revised version of our manuscript may now be found suitable for publication.

Sincerely,

Rossella Cianci

Reviewer 2 Report

Comments and Suggestions for Authors

The potential role of different microbiota could perhaps be listed or demonstrated also in tabular form. 

Author Response

Rome, 14th November, 2023

Dear Editor of “Cells”,

first of all, my coauthors and I would like to thank You sincerely for this opportunity of cooperation, following the submission of the paper “Gut microbiota, inflammatory bowel disease and cancer: the role of guardians of innate immunity” and its possible publication upon “Cells”.

We profoundly thank the reviewer for the comments and useful suggestions aimed at improving the final version of the paper.

This is a point-by-point list of changes made in the paper:

Reviewer #2

The potential role of different microbiota could perhaps be listed or demonstrated also in tabular form.

We have added a table, as suggested.

We thank You for your constructive critique and we hope the review process has led to an improved manuscript.

If additional changes are warranted, we will make them.

We hope that this revised version of our manuscript may now be found suitable for publication.

Sincerely,

Rossella Cianci

Reviewer 3 Report

Comments and Suggestions for Authors

A nicely written review on the association between the innate immune response and specifically PRRs and their down-stream pathways to the gut microbiome, IBD and cancer.

Below are my comments:

1.      Although I have not looked at all the references but by sampling some of them it seems that- Reference 27 is not relevant to the text and it is hard to say that reference 28 is a good enough reference, as the reference is anecdotal. Please make sure that all references are relevant and are aligned with the text.  

2.      Table 1: I suggest adding to the Table some data- first of all, each statement should be accompanied by the relevant reference. Secondly, it seems to me that most of the associations between IBD, or cancer and TLR mutations are based on very few cases. Please clarify this in the text, and perhaps add to the table the level of evidence and whether the study is based on in-vitro data, animal models or humans.

3.      The statement regarding the protective effects of F prausnitzii against cancer is not clear-cut as appears in the review reference provided by the authors [REF- 97], where other studies have reported no depletion in F prausnitzii levels in CRC (Balamurugan et al., 2008; Sobhani et al., 2011; Wang et al., 2012).

4.      How come other bacteria such Fusobacterium nucleatum are not mentioned in the context of fecal microbiome and CRC?

5.      Line 440-441- regarding TLR-9 and cancer. Should be mentioned that the study is based on a mouse model [REF 103]. In any case, even in that study, it is mentioned that "the findings may be controversial because other previous studies have revealed that TLR9 agonists exerted an antitumor effect in CRC (14,15,44,45)."

This is one of the major weaknesses of the current review. It presents the data in very definite way, despite data that are some times based on in-vitro data, animal experiments or just case reports. The authors should be transparent and mention the level of strengths of the statements.

6.      There is no integration of the data through an example that shows the complete suggested mechanism of altered bacterial relative ratio in the context of IBD, relation to TLR activation/expression/ mutation, and relation to cancer. I am not sure there are good publications in this regard. One study that should probably be cited, looks at TLRs, specific bacteria and CRC, is- "Expression of Main Toll-Like Receptors in Patients with Different Types of Colorectal Polyps and Their Relationship with Gut Microbiota.Rezasoltani S, Ghanbari R, Looha MA, Mojarad EN, Yadegar A, Stewart D, Aghdaei HA, Zali MR.Int J Mol Sci. 2020 Nov 26;21(23):8968. doi: 10.3390/ijms21238968. "

But other studies should be looked for, especially in the context of IBD and in any case, this should be discussed.  

Comments on the Quality of English Language

Typos/ Grammar-

Table 1:

TLR4- should be "causes tissue destruction"

TLR4- should be "TLR4 deficient mice"

Line 436- should "on the one hand"

Author Response

Rome, 14th November, 2023

Dear Editor of “Cells”,

first of all, my coauthors and I would like to thank You sincerely for this opportunity of cooperation, following the submission of the paper “Gut microbiota, inflammatory bowel disease and cancer: the role of guardians of innate immunity” and its possible publication upon “Cells”.

We profoundly thank the reviewer for the comments and useful suggestions aimed at improving the final version of the paper.

This is a point-by-point list of changes made in the paper:

Reviewer #3

A nicely written review on the association between the innate immune response and specifically PRRs and their down-stream pathways to the gut microbiome, IBD and cancer. Below are my comments:

  1. Although I have not looked at all the references but by sampling some of them it seems that- Reference 27 is not relevant to the text and it is hard to say that reference 28 is a good enough reference, as the reference is anecdotal. Please make sure that all references are relevant and are aligned with the text.

We have modified several references, as suggested

  1. Table 1: I suggest adding to the Table some data- first of all, each statement should be accompanied by the relevant reference. Secondly, it seems to me that most of the associations between IBD, or cancer and TLR mutations are based on very few cases. Please clarify this in the text, and perhaps add to the table the level of evidence and whether the study is based on in-vitro data, animal models or humans.

Thanks for the comment. We have added the requested data in Table 1.

  1. The statement regarding the protective effects of F prausnitzii against cancer is not clear-cut as appears in the review reference provided by the authors [REF- 97], where other studies have reported no depletion in F prausnitzii levels in CRC (Balamurugan et al., 2008; Sobhani et al., 2011; Wang et al., 2012).

We have modified the text, as follows: 'However, the protective effects of F. prausnitzii are not clear cut-defined since other studies have shown no depletion of the microorganism in CRC (https://doi.org/10.1371/journal.pone.0016393; DOI: 10.1038/ismej.2011.109)’.

  1. How come other bacteria such Fusobacterium nucleatum are not mentioned in the context of fecal microbiome and CRC?

We have added the role of F. nucleatum, as suggested. ‘Fusobacterium nucleatum is suggested as another predictive and prognostic biomarker for CRC. In particular, the intratumoral presence of this CRC-enriching microorganism correlates with a poor prognosis due to higher microsatellite instability and gene mutation (https://doi.org/10.1158/0008-5472.CAN-13-1865). Different experimental models demonstrated that Fusobacterium nucleatum promotes tumor progression, metastatization and chemoresistance through its ability to influence tumor cells and several tumor microenvironment components (i.e. extracellular matrix, immune cell and stromal cells). However, also the association between Fusobacterium nucleatum and CRC initiation is still unresolved and needs to be further elucidated (doi: 10.3389/fcell.2021.710165). Fusobacterium nucleatum cannot be completely considered as a pro-carcinogenic microorganism since its role depends upon the genetic background of the host, tumor microenvironment and environmental factors’.

  1. Line 440-441- regarding TLR-9 and cancer. Should be mentioned that the study is based on a mouse model [REF 103]. In any case, even in that study, it is mentioned that "the findings may be controversial because other previous studies have revealed that TLR9 agonists exerted an antitumor effect in CRC (14,15,44,45)."

This is one of the major weaknesses of the current review. It presents the data in very definite way, despite data that are some times based on in-vitro data, animal experiments or just case reports. The authors should be transparent and mention the level of strengths of the statements.

We thank the reviewer for pointing this out. We have now mentioned that those findings are still controversial and reduced the strength level of previous statement.

  1. There is no integration of the data through an example that shows the complete suggested mechanism of altered bacterial relative ratio in the context of IBD, relation to TLR activation/expression/ mutation, and relation to cancer. I am not sure there are good publications in this regard. One study that should probably be cited, looks at TLRs, specific bacteria and CRC, is- "Expression of Main Toll-Like Receptors in Patients with Different Types of Colorectal Polyps and Their Relationship with Gut Microbiota.Rezasoltani S, Ghanbari R, Looha MA, Mojarad EN, Yadegar A, Stewart D, Aghdaei HA, Zali MR.Int J Mol Sci. 2020 Nov 26;21(23):8968. doi: 10.3390/ijms21238968. "

But other studies should be looked for, especially in the context of IBD, and in any case, this should be discussed.

We thank the reviewer for this suggestion. We have now mentioned the work of Rezasoltani S, et al., Int J Mol Sci. 2020. We have also included other works to highlight the role of TLR immune signaling induced by microbiota in cancer progression and response to different conventional treatments.

Typos/ Grammar-

Table 1:

TLR4- should be "causes tissue destruction"

TLR4- should be "TLR4 deficient mice"

Line 436- should "on the one hand"

We have modified typos throughout the text

We thank You for your constructive critique and we hope the review process has led to an improved manuscript.

If additional changes are warranted, we will make them.

We hope that this revised version of our manuscript may now be found suitable for publication.

Sincerely,

Rossella Cianci

Round 2

Reviewer 1 Report

Comments and Suggestions for Authors

The author has responded to the reviewer's comment point by point. It can be accepted in the current revision.

Comments on the Quality of English Language

The author has responded to the reviewer's comment point by point. It can be accepted in the current revision.